# Constructive Universal Approximation Theorems for Deep Joint-Equivariant Networks by Schur's Lemma

## Abstract

We present a unified constructive universal approximation theorem covering a wide range of learning machines including both shallow and deep neural networks based on the group representation theory. Constructive here means that the distribution of parameters is given in a closed-form expression (called the *ridgelet transform*). Contrary to the case of shallow models, expressive power analysis of deep models has been conducted in a case-by-case manner. Recently, Sonoda et al. [33, 32] developed a systematic method to show a constructive approximation theorem from *scalar-valued joint-group-invariant* feature maps, covering a formal deep network. However, each hidden layer was formalized as an abstract group action, so it was not possible to cover real deep networks defined by composites of nonlinear activation function. In this study, we extend the method for *vector-valued joint-group-equivariant* feature maps, so to cover such real networks.

## 1 Introduction

An ultimate goal of the deep learning theory is to characterize the internal data processing procedure inside deep neural networks obtained by deep learning. We may formulate this problem as a functional equation problem: Let $\mathcal{F}$ denote a class of data generating functions, and let $\mathtt{DNN}[\gamma]$ denote a certain deep neural network with parameter $\gamma$. Given a function $f \in \mathcal{F}$, find an unknown parameter $\gamma$ so that network $\mathtt{DNN}[\gamma]$ represents function $f$, i.e.

$$\mathtt{DNN}[\gamma] = f, \tag{1}$$

which we call a *DNN equation*. An ordinary learning problem by empirical risk minimization, such as minimizing $\sum_{i=1}^{n} |\mathtt{DNN}[\gamma](x_i) - f(x_i)|^2$ with respect to $\gamma$, is understood as a weak form (or a variational form) of this equation. Therefore, characterizing the solution space of this equation leads to understanding the parameters obtained by deep learning. Following previous studies [21, 3, 28–31], we call a solution operator $\mathtt{R}$ that satisfies $\mathtt{DNN}[\mathtt{R}[f]] = f$ a *ridgelet transform*. Once such a solution operator $\mathtt{R}$ is found, we can conclude a *universality* of the DNN in consideration because the reconstruction formula $\mathtt{DNN}[\mathtt{R}[f]] = f$ implies for any $f \in \mathcal{F}$ there exists a DNN that represents $f$. In particular, when $\mathtt{R}[f]$ is found in a closed-form manner, then it leads to a *constructive* proof of the universality since $\mathtt{R}[f]$ could indicate how to assign parameters.

When the network has only one infinitely-wide hidden layer, though it is not deep but shallow, the characterization problem has been well investigated. For example, the learning dynamics and the global convergence property (of SGD) are well studied in the mean field theory [22, 25, 20, 5] and the Langevin dynamics theory [35], and even closed-form solution operator to a "shallow" NN equation, the original ridgelet transform, has already been presented [28–31].

On the other hand, when the network has more than one hidden layer, the problem is far from solved, and it is common to either consider infinitely-deep mathematical models such as Neural

ODEs [27, 9, 17, 12, 4], or handcraft inner feature maps depending on the problem. For example, construction methods such as the Telgarsky sawtooth function (or the Yarotsky scheme) and bit extraction techniques [7, 36–39, 8, 6, 26, 24, 11] have been developed to demonstrate the depth separation, super-convergence, and minmax optimality of deep ReLU networks. Various feature maps have also been handcrafted in the contexts of geometric deep learning [1] and deep narrow networks [19, 13, 18, 14, 23, 16, 2, 15]. Needless to say, there is no guarantee that these handcrafted feature maps are acquired by deep learning, so these analyses are considered to be analyses of possible worlds.

Recently, Sonoda et al. [33, 32] discovered a rich class of ridgelet transforms for learning machines defined by *scalar-valued joint-group-invariant* feature maps, covering both depth-2 fully-connected networks and the formal deep network (FDN), yielding the first ridgelet transform for deep models. Their theory is indeed a breakthrough because it could cover both deep and shallow models simultaneously. However, each hidden layer in the FDN has to be formalized as an abstract group action, so it was not possible to cover real deep networks defined by composites of nonlinear activation function. In this study, we extend their arguments for *vector-valued joint-group-equivariant* feature maps (Theorem 3 and Corollary 1), so to cover such real networks. As an important example, in § 4.2, we obtained the ridgelet transform for a more realistic DNN, the depth-$n$ fully-connected network with an arbitrary activation function (not limited to ReLU), without handcrafting network architecture. In other words, it is a constructive proof of the $L^2(\mathbb{R}^m; \mathbb{R}^m)$-universality of the DNNs, and an explicit characterization of the solution space of the DNN equation for more realistic setup.

Thanks to Schur's lemma, a basic and useful result in the representation theory, the proof of the main theorem is surprisingly simple, yet the scope of application is wide. The significance of this study lies in revealing the close relationship between machine learning theory and modern algebra. With this study as a catalyst, we expect a major upgrade to machine learning theory from the perspective of modern algebra.

# 2  Preliminaries

We quickly introduce the original integral representation and the ridgelet transform, a mathematical model of depth-2 fully-connected network and its right inverse. Then, we list a few facts in the group representation theory. In particular, *Schur's lemma* and the *Haar measure* play key roles in the proof of the main results.

**Notation.**  For any topological space $X$, $C_c(X)$ denotes the Banach space of all compactly supported continuous functions on $X$. For any measure space $X$, $L^p(X)$ denotes the Banach space of all $p$-integrable functions on $X$. $\mathcal{S}(\mathbb{R}^d)$ and $\mathcal{S}'(\mathbb{R}^d)$ denote the classes of rapidly decreasing functions (or Schwartz test functions) and tempered distributions on $\mathbb{R}^d$, respectively.

## 2.1  Integral Representation and Ridgelet Transform for Depth-2 Fully-Connected Network

**Definition 1.**  For any measurable functions $\sigma : \mathbb{R} \to \mathbb{C}$ and $\gamma : \mathbb{R}^m \times \mathbb{R} \to \mathbb{C}$, put

$$S_\sigma[\gamma](\boldsymbol{x}) := \int_{\mathbb{R}^m \times \mathbb{R}} \gamma(\boldsymbol{a}, b)\sigma(\boldsymbol{a} \cdot \boldsymbol{x} - b)\mathrm{d}\boldsymbol{a}\mathrm{d}b, \quad \boldsymbol{x} \in \mathbb{R}^m. \tag{2}$$

We call $S_\sigma[\gamma]$ an (integral representation of) neural network, and $\gamma$ a parameter distribution.

The integration over all the hidden parameters $(\boldsymbol{a}, b) \in \mathbb{R}^m \times \mathbb{R}$ means all the neurons $\{\boldsymbol{x} \mapsto \sigma(\boldsymbol{a} \cdot \boldsymbol{x} - b) \mid (\boldsymbol{a}, b) \in \mathbb{R}^m \times \mathbb{R}\}$ are summed (or integrated, to be precise) with weight $\gamma$, hence formally $S_\sigma[\gamma]$ is understood as a continuous neural network with a single hidden layer. We note, however, when $\gamma$ is a finite sum of point measures such as $\gamma_p = \sum_{i=1}^p c_i \delta_{(\boldsymbol{a}_i, b_i)}$ (by appropriately extending the class of $\gamma$ to Borel measures), then it can also reproduce a finite width network

$$S_\sigma[\gamma_p](\boldsymbol{x}) = \sum_{i=1}^p c_i \sigma(\boldsymbol{a}_i \cdot \boldsymbol{x} - b_i). \tag{3}$$

In other words, the integral representation is a mathmatical model of depth-2 network with *any* width (ranging from finite to continuous).

Next, we introduce the ridgelet transform, which is known to be a right-inverse operator to $S_\sigma$.

**Definition 2.** For any measurable functions $\rho : \mathbb{R} \to \mathbb{C}$ and $f : \mathbb{R}^m \to \mathbb{C}$, put

$$R_\rho[f](\boldsymbol{a}, b) := \int_{\mathbb{R}^m} f(\boldsymbol{x})\overline{\rho(\boldsymbol{a} \cdot \boldsymbol{x} - b)}\mathrm{d}\boldsymbol{x}, \quad (\boldsymbol{a}, b) \in \mathbb{R}^m \times \mathbb{R}. \tag{4}$$

We call $R_\rho$ a ridgelet transform.

To be precise, it satisfies the following reconstruction formula.

**Theorem 1** (Reconstruction Formula). *Suppose $\sigma$ and $\rho$ are a tempered distribution ($\mathcal{S}'$) and a rapid decreasing function ($\mathcal{S}$) respectively. There exists a bilinear form $((\sigma, \rho))$ such that*

$$S_\sigma \circ R_\rho[f] = ((\sigma, \rho))f, \tag{5}$$

*for any square integrable function $f \in L^2(\mathbb{R}^m)$. Further, the bilinear form is given by $((\sigma, \rho)) = \int_{\mathbb{R}} \sigma^\sharp(\omega)\overline{\rho^\sharp(\omega)}|\omega|^{-m}\mathrm{d}\omega$, where $\sharp$ denotes the 1-dimensional Fourier transform.*

See Sonoda et al. [29, Theorem 6] for the proof. In particular, according to Sonoda et al. [29, Lemma 9], for any activation function $\sigma$, there always exists $\rho$ satisfying $((\sigma, \rho)) = 1$. Here, $\sigma$ being a tempered distribution means that typical activation functions are covered such as ReLU, step function, $\tanh$, gaussian, etc... We can interpret the reconstruction formula as a universality theorem of continuous neural networks, since for any given data generating function $f$, a network with output weight $\gamma_f = R_\rho[f]$ reproduces $f$ (up to factor $((\sigma, \rho))$), i.e. $S[\gamma_f] = f$. In other words, the ridgelet transform indicates how the network parameters should be organized so that the network represents an individual function $f$.

The original ridgelet transform was discovered by Murata [21] and Candès [3]. It is recently extended to a few modern networks by the Fourier slice method [34, see e.g.]. In this study, we present a systematic scheme to find the ridgelet transform for a variety of given network architecture based on the group theoretic arguments.

## 2.2 Irreducible Unitary Representation and Schur's Lemma

Let $G$ be a locally compact group, $\mathcal{H}$ be a nonzero Hilbert space, and $\mathcal{U}(\mathcal{H})$ be the group of unitary operators on $\mathcal{H}$. For example, any finite group, discrete group, compact group, and finite-dimensional Lie group are locally compact, while an infinite-dimensional Lie group is not locally compact. A *unitary representation* $\pi$ of $G$ on $\mathcal{H}$ is a group homomorphism that is continuous with respect to the strong operator topology—that is, a map $\pi : G \to \mathcal{U}(\mathcal{H})$ satisfying $\pi(gh) = \pi(g)\pi(h)$ and $\pi(g^{-1}) = \pi(g)^{-1}$, and for any $\psi \in \mathcal{H}$, the map $G \ni g \mapsto \pi(g)[\psi] \in \mathcal{H}$ is continuous.

Suppose $\mathcal{M}$ is a closed subspace of $\mathcal{H}$. $\mathcal{M}$ is called an *invariant* subspace when $\pi(g)\mathcal{M} \subset \mathcal{M}$ for all $g \in G$. Particularly, $\pi$ is called *irreducible* when it does not admit any nontrivial invariant subspace $\mathcal{M} \neq \{0\}$ nor $\mathcal{H}$. The following theorem is a fundamental result of group representation theory that characterizes the irreducibility.

**Theorem 2** (Schur's lemma). *A unitary representation $(\pi, \mathcal{H})$ is irreducible iff any bounded operator $T$ on $\mathcal{H}$ that commutes with $\pi$ is always a constant multiple of the identity. In other words, if $\pi(g)T = T\pi(g)$ for all $g \in G$, then $T = c\,\mathrm{Id}_\mathcal{H}$ for some $c \in \mathbb{C}$.*

See Folland [10, Theorem 3.5(a)] for the proof. We use this as a key step in the proof of our main theorem.

## 2.3 Calculus on Locally Compact Group

By Haar's theorem, if $G$ is a locally compact group, then there uniquely exist left and right invariant measures $\mathrm{d}_l g$ and $\mathrm{d}_r g$, satisfying for any $s \in G$ and $f \in C_c(G)$,

$$\int_G f(sg)\mathrm{d}_l g = \int_G f(g)\mathrm{d}_l g, \quad \text{and} \quad \int_G f(gs)\mathrm{d}_r g = \int_G f(g)\mathrm{d}_r g.$$

Let $X$ be a $G$-space with transitive left (resp. right) $G$-action $g \cdot x$ (resp. $x \cdot g$) for any $(g, x) \in G \times X$. Then, we can further induce the left (resp. right) invariant measure $\mathrm{d}_l x$ (resp. $\mathrm{d}_r x$) so that for any $f \in C_c(G)$,

$$\int_X f(x)\mathrm{d}_l x := \int_G f(g \cdot o)\mathrm{d}_l g, \quad \text{resp.} \quad \int_X f(x)\mathrm{d}_r x := \int_G f(o \cdot g)\mathrm{d}_r g,$$

where $o \in X$ is a fixed point called the origin.

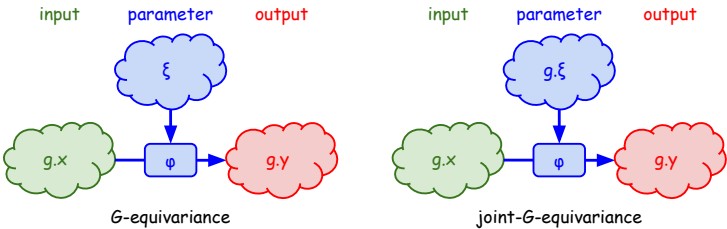

Figure 1: An ordinary $G$-equivariant feature map $\phi : X \times \Xi \to Y$ is a subclass of joint-$G$-equivariant map where the $G$-action on parameter domain $\Xi$ is *trivial*, i.e. $g \cdot \xi = \xi$

## 3    Main Results

We introduce the joint-group-equivariant feature map, and present the ridgelet transforms for learning machines defined by joint-group-equivariant feature maps, yielding the universality of deep models.

Let $G$ be a locally compact group equipped with a left invariant measure $\mathrm{d}g$. Let $X$ and $\Xi$ be $G$-spaces equipped with $G$-invariant measures $\mathrm{d}x$ and $\mathrm{d}\xi$, called the data domain and the parameter domain, respectively. Particularly, we call the product space $X \times \Xi$ the *data-parameter* domain (like time-frequency domain), and call any map $\phi$ on data-parameter domain $X \times \Xi$ a *feature map*. Let $\mathcal{H}$ be a separable Hilbert space, let $\mathcal{U}(\mathcal{H})$ be the space of unitary operators on $\mathcal{H}$, and let $\upsilon : G \to \mathcal{U}(\mathcal{H})$ be a unitary representation of $G$ on $\mathcal{H}$. If there is no danger of confusion, we use the same symbol $\cdot$ for the $G$-actions on $X$, $\mathcal{H}$, and $\Xi$ (e.g., $g \cdot x$, $g \cdot \upsilon$, and $g \cdot \xi$).

In the main theorem, the irreducibility of the following unitary representation $\pi$ will be a sufficient condition for the universality. Let $L^2(X; \mathcal{H})$ denote the space of $\mathcal{H}$-valued square-integrable functions on $X$ equipped with the inner product $\langle \phi, \psi \rangle_{L^2(X;\mathcal{H})} := \int_X \langle \phi(x), \psi(x) \rangle_{\mathcal{H}} \mathrm{d}x$. Put

$$\pi_g[f](x) := g \cdot f(g^{-1} \cdot x), \quad x \in X, \ f \in L^2(X; \mathcal{H}), \ g \in G. \tag{6}$$

Then, it is a unitary representation of $G$ on $L^2(X; \mathcal{H})$. In fact, $\pi_g[\pi_h[f]](x) = g \cdot h \cdot f(h^{-1} \cdot g^{-1} \cdot x) = (gh) \cdot f((gh)^{-1} \cdot x) = \pi_{gh}[f](x)$, and $\langle \pi_g[f_1], \pi_g[f_2] \rangle_{L^2(X;\mathcal{H})} = \int_X \langle \upsilon_g[f_1](g^{-1} \cdot x), \upsilon_g[f_2](g^{-1} \cdot x) \rangle_{\mathcal{H}} \mathrm{d}x = \int_X \langle f_1(x), \upsilon_g^*[\upsilon_g[f_2]](x) \rangle_{\mathcal{H}} \mathrm{d}x = \langle f_1, f_2 \rangle_{L^2(X;\mathcal{H})}$.

In addition, let $L^2(\Xi)$ denote the space of $\mathbb{C}$-valued square-integrable functions on $\Xi$, and let $\widehat{\pi}$ be the left-regular representation of $G$ on $L^2(\Xi)$ given by

$$\widehat{\pi}_g[\gamma](\xi) := \gamma(g^{-1} \cdot \xi), \quad \xi \in \Xi, \ \gamma \in L^2(\Xi), \ g \in G. \tag{7}$$

Similarly to $\pi$, $\widehat{\pi}$ is also a unitary representation.

**Definition 3** (Joint $G$-Equivariant Feature Map). Let $X, Y$ be data domains, and $\Xi$ be a parameter domain (with $G$-actions). We say a feature map $\phi : X \times \Xi \to Y$ is *joint-$G$-equivariant* when

$$\phi(g \cdot x, g \cdot \xi) = g \cdot \phi(x, \xi), \quad (x, \xi) \in X \times \Xi, \tag{8}$$

holds for all $g \in G$. In other words, $\phi$ is a homomorphism (or $G$-map) of $G$-sets from $X \times \Xi$ to $Y$. So by $\hom_G(X \times \Xi, Y)$, we denote the collection of all joint-$G$-equivariant maps. Additionally, when $G$-action on $Y$ is trivial, i.e. $\phi(g \cdot x, g \cdot \xi) = \phi(x, \xi)$, we say it is *joint-$G$-invariant*.

*Remark* 1. The joint-$G$-equivariance extends an ordinary notion of $G$-*equivariance*, i.e. $\phi(g \cdot x, \xi) = g \cdot \phi(x, \xi)$. In fact, $G$-equivariance is a special case of joint-$G$-equivariance where $G$ acts trivially on parameter domain, i.e. $g \cdot \xi = \xi$ (see also Figure 1).

In order to construct a (non-joint) group-equivariant network, we must carefully and precisely design the network architecture [see, e.g., a textbook of geometric deep learning 1]. On the other hand, we can easily and systematically construct joint-$G$-equivariant network from (not at all equivariant but) *any* map $f : X \to Y$ according to the following Lemmas 1 and 2.

**Lemma 1.** *Suppose group $G$ acts on sets $X$ and $Y$. Fix an arbitrary map $f : X \to Y$, and put $\phi(x, g) := g \cdot f(g^{-1} \cdot x)$ for every $x \in X$ and $g \in G$. Then, $\phi : X \times G \to Y$ is joint-$G$-equivariant.*

*Proof.* Straightforward. For any $g \in G$, $\phi(g \cdot x, g \cdot h) = (gh) \cdot f((gh)^{-1} \cdot (g \cdot x)) = g \cdot \phi(x, h)$. $\quad\square$

**Lemma 2** (Depth-$n$ Feature Map $\phi_{1:n}$)**.** *Given a sequence of $G$-equivariant feature maps $\phi_i :$ $X_{i-1} \times \Xi_i \to X_i$ $(i = 1, \ldots, n)$, put $\phi_{1:n} : X_0 \times \Xi_1 \times \cdots \times \Xi_n \to X_n$ by*

$$\phi_{1:n}(x, \xi_1, \ldots, \xi_n) := \phi_n(\bullet, \xi_n) \circ \cdots \circ \phi_1(x, \xi_1). \tag{9}$$

*Then, $\phi_{1:n}$ is $G$-equivariant. Following the custom of counting the number of parameter domains $(\Xi_i)_{i=1}^{n}$, we say $\phi_{1:n}$ is depth-$n$.*

*Proof.* In fact,

$$
\begin{aligned}
\phi_{1:n}(g \cdot x, g \cdot \xi_1, \ldots, g \cdot \xi_n) &= \phi_n(\bullet, g \cdot \xi_n) \circ \cdots \circ \phi_2(\bullet, g \cdot \xi_2) \circ \phi_1(g \cdot x, g \cdot \xi_1) \\
&= \phi_n(\bullet, g \cdot \xi_n) \circ \cdots \circ \phi_2(g \cdot \bullet, g \cdot \xi_2) \circ \phi_1(x, \xi_1) \\
&\quad\vdots \\
&= \phi_n(g \cdot \bullet, g \cdot \xi_n) \circ \cdots \circ \phi_2(\bullet, \xi_2) \circ \phi_1(x, \xi_1) \\
&= g \cdot \phi_n(\bullet, \xi_n) \circ \cdots \circ \phi_2(\bullet, \xi_2) \circ \phi_1(x, \xi_1) \\
&= g \cdot \phi_{1:n}(x, \xi_1, \ldots, \xi_n). \qquad\qquad \square
\end{aligned}
$$

**Definition 4** ($\phi$-Network)**.** For any vector-valued map $\phi : X \times \Xi \to \mathcal{H}$ and scalar-valued map $\gamma : \Xi \to \mathbb{C}$, define a vector-valued map $X \to \mathcal{H}$ by

$$\mathtt{NN}[\gamma; \phi](x) := \int_{\Xi} \gamma(\xi)\phi(x, \xi)\mathrm{d}\xi, \quad x \in X, \tag{10}$$

where the integral is understood as the Bocher integral.

We call the integral transform $\mathtt{NN}[\bullet; \phi]$ a $\phi$-transform, and each individual image $\mathtt{NN}[\gamma; \phi]$ a $\phi$-network for short. The $\phi$-network extends the original integral representation. In particular, it inherits the concept of integrating all the possible parameters $\xi$ and indirectly select which parameters to use by weighting on them, which *linearize* parametrization by lifting nonlinear parameters $\xi$ to linear parameter $\gamma$.

**Definition 5** ($\psi$-Ridgelet Transform)**.** For any $\mathcal{H}$-valued feature map $\psi : X \times \Xi \to \mathcal{H}$ and $\mathcal{H}$-valued Borel measurable function $f$ on $X$, put a scalar-valued integral transform

$$\mathtt{R}[f; \psi](\xi) := \int_X \langle f(x), \psi(x, \xi) \rangle_{\mathcal{H}} \mathrm{d}x, \quad \xi \in \Xi. \tag{11}$$

We call the integral transform $\mathtt{R}[\bullet; \psi]$ a $\psi$-ridgelet transform for short.

As long as the integrals are convergent, $\phi$-ridgelet transform is the dual operator of $\phi$-transform, since

$$\langle \gamma, \mathtt{R}[f; \phi] \rangle_{L^2(\Xi)} = \int_{X \times \Xi} \gamma(\xi) \langle \phi(x, \xi), f(x) \rangle_{\mathcal{H}} \mathrm{d}x\mathrm{d}\xi = \langle \mathtt{NN}[\gamma; \phi], f \rangle_{L^2(X; \mathcal{H})}. \tag{12}$$

**Theorem 3** (Reconstruction Formula)**.** *Assume (1) $\mathcal{H}$-valued feature maps $\phi, \psi : X \times \Xi \to \mathcal{H}$ are joint-$G$-equivariant, (2) composite operator $\mathtt{NN}_\phi \circ \mathtt{R}_\psi : L^2(X; \mathcal{H}) \to L^2(X; \mathcal{H})$ is bounded (i.e., Lipschitz continuous), and (3) the unitary representation $\pi$ defined in (6) is irreducible. Then, there exists a bilinear form $((\phi, \psi)) \in \mathbb{C}$ (independent of $f$) such that for any $\mathcal{H}$-valued square-integrable function $f \in L^2(X; \mathcal{H})$,*

$$\mathtt{NN}_\phi \circ \mathtt{R}_\psi[f] = ((\phi, \psi))f. \tag{13}$$

In other words, the $\psi$-ridgelet transform $\mathtt{R}_\psi$ is a right inverse operator of $\phi$-transform $\mathtt{NN}_\phi$ as long as $((\phi, \psi)) \neq 0, \infty$.

*Proof.* We write $\mathtt{NN}[\bullet; \phi]$ as $\mathtt{NN}_\phi$ and $\mathtt{R}[\bullet; \phi]$ as $\mathtt{R}_\phi$ for short. By using the unitarity of representation $\upsilon : G \to \mathcal{U}(\mathcal{H})$, left-invariance of measure $\mathrm{d}x$, and $G$-equivariance of feature map $\psi$, for all $g \in G$, we have

$$
\begin{aligned}
\mathtt{R}_\psi[\pi_g[f]](\xi) &= \int_X \langle g \cdot f(g^{-1} \cdot x), \psi(x, \xi) \rangle_{\mathcal{H}} \mathrm{d}x = \int_X \langle f(x), g^{-1} \cdot \psi(g \cdot x, \xi) \rangle_{\mathcal{H}} \mathrm{d}x \\
&= \int_X \langle f(x), \psi(x, g^{-1} \cdot \xi) \rangle_{\mathcal{H}} \mathrm{d}x = \widehat{\pi}_g[\mathtt{R}_\psi[f]](\xi). \tag{14}
\end{aligned}
$$

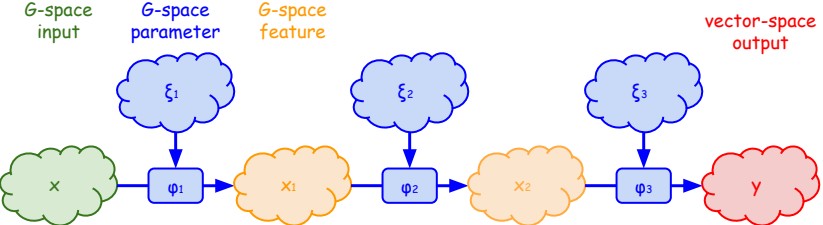

Figure 2: Deep $\mathcal{H}$-valued joint-$G$-equivariant network on $G$-space $X$ is $L^2(X;\mathcal{H})$-universal when unitary representation $\pi$ of $G$ on $L^2(X;\mathcal{H})$ is irreducible, and the distribution of parameters for the network to represent a given map $f : X \to \mathcal{H}$ is exactly given by the ridgelet transform $\mathtt{R}[f]$

Similarly,

$$\mathtt{NN}_\phi[\widehat{\pi}_g[\gamma]](x) = \int_\Xi \gamma(g^{-1} \cdot \xi)\phi(x,\xi)\mathrm{d}\xi = \int_\Xi \gamma(\xi)\phi(x, g \cdot \xi)\mathrm{d}\xi$$

$$= \int_\Xi \gamma(\xi) \left( g \cdot \phi(g^{-1} \cdot x, \xi) \right) \mathrm{d}\xi = \pi_g[\mathtt{NN}_\phi[\gamma]](x). \tag{15}$$

Here, $\widehat{\pi}^*$ denotes the dual representation of $\widehat{\pi}$ with respect to $L^2(\Xi)$.

As a consequence, $\mathtt{NN}_\phi \circ \mathtt{R}_\psi : L^2(X;\mathcal{H}) \to L^2(X;\mathcal{H})$ commutes with $\pi$ as below

$$\mathtt{NN}_\phi \circ \mathtt{R}_\psi \circ \pi_g = \mathtt{NN}_\phi \circ \widehat{\pi}_g \circ \mathtt{R}_\psi = \pi_g \circ \mathtt{NN}_\phi \circ \mathtt{R}_\psi \tag{16}$$

for all $g \in G$. Hence by Schur's lemma (Theorem 2), there exist a constant $C_{\phi,\psi} \in \mathbb{C}$ such that $\mathtt{NN}_\phi \circ \mathtt{R}_\psi = C_{\phi,\psi} \, \mathrm{Id}_{L^2(X)}$. Since $\mathtt{NN}_\phi \circ \mathtt{R}_\psi$ is bilinear in $\phi$ and $\psi$, $C_{\phi,\psi}$ is bilinear in $\phi$ and $\psi$. $\qquad\square$

In particular, because depth-$n$ feature map $\phi_{1:n}$ is $G$-equivariant (Lemma 2), the following depth-$n$ $\mathcal{H}$-valued deep network $\mathtt{DNN}[\gamma; \phi_{1:n}]$ is $L^2(X;\mathcal{H})$-universal.

**Corollary 1** (Deep Ridgelet Transform). *For any maps $\gamma : X \to \mathbb{C}$ and $f \in L^2(X;\mathcal{H})$, put*

$$\mathtt{DNN}[\gamma; \phi_{1:n}](x) := \int_{\Xi_1 \times \cdots \times \Xi_n} \gamma(\xi_1, \ldots, \xi_n)\phi_n(\bullet, \xi_n) \circ \cdots \circ \phi_1(x, \xi_1)\mathrm{d}\boldsymbol{\xi}, \quad x \in X, \tag{17}$$

$$\mathtt{R}[f; \psi_{1:n}](\boldsymbol{\xi}) := \int_\Xi \langle f(x), \psi_n(\bullet, \xi_n) \circ \cdots \circ \psi_1(x, \xi_n) \rangle_{\mathcal{H}} \mathrm{d}x, \quad \boldsymbol{\xi} \in \Xi_1 \times \cdots \times \Xi_n. \tag{18}$$

*Under the assumptions that $\mathtt{DNN}_{\phi_{1:n}} \circ \mathtt{R}_{\psi_{1:n}}$ is bounded, and that $\pi$ is irreducible, there exists a bilinear form $((\phi_{1:n}, \psi_{1:n}))$ satisfying $\mathtt{DNN}_{\phi_{1:n}} \circ \mathtt{R}_{\psi_{1:n}} = ((\phi_{1:n}, \psi_{1:n})) \, \mathrm{Id}_{L^2(X;\mathcal{H})}$.*

Again, it extends the original integral representation, and inherits the *linearization* trick of nonlinear parameters $\boldsymbol{\xi}$ by integrating all the possible parameters (beyond the difference of layers) and indirectly select which parameters to use by weighting on them.

## 4 Example: Depth-$n$ Fully-Connected Network with Arbitrary Activation

As a concrete example, we present the ridgelet transform for depth-$n$ fully-connected network. First, we show the depth-2 case based on a joint-affine-*invariant* argument, which was originally demonstrated by Sonoda et al. [33]. Then, we show the depth-$n$ case based on a joint-*equivariant* argument by extending the original arguments.

We use the following known facts.

**Lemma 3.** *The regular representation $\pi$ of the affine group $\mathrm{Aff}(m)$ on $L^2(\mathbb{R}^m)$ (defined below) is irreducible.*

See Folland [10, Theorem 6.42] for the proof.

**Lemma 4.** *Suppose $\sigma$ and $\rho$ are a tempered distribution $(\mathcal{S}')$ and a Schwartz test function, respectively. Then, $S_\sigma \circ R_\rho : L^2(\mathbb{R}^m) \to L^2(\mathbb{R}^m)$ is bounded.*

See Sonoda et al. [29, Lemmas 7 and 8] for the proof.

## 4.1 Depth-2

Set $X := \mathbb{R}^m$ (data domain), $\Xi := \mathbb{R}^m \times \mathbb{R}$ (parameter domain), and $G := \mathrm{Aff}(m) = GL(m) \ltimes \mathbb{R}^m$ be the $m$-dimensional affine group, acting on data domain $X$ by

$$g \cdot \boldsymbol{x} := L\boldsymbol{x} + \boldsymbol{t}, \quad g = (L, \boldsymbol{t}) \in GL(m) \ltimes \mathbb{R}^m, \ \boldsymbol{x} \in X. \tag{19}$$

Addition to this, let $\pi$ be the regular representation of $\mathrm{Aff}(m)$ on $L^2(X)$, namely

$$\pi(g)[f](\boldsymbol{x}) := |\det L|^{-1/2} f(L^{-1}(\boldsymbol{x} - \boldsymbol{t})), \quad f \in L^2(X) \text{ and } g = (L, \boldsymbol{t}) \in GL(m) \ltimes \mathbb{R}^m. \tag{20}$$

Further, define a *dual action* of $\mathrm{Aff}(m)$ on the parameter domain $\Xi$ as

$$g \cdot (\boldsymbol{a}, b) = (L^{-\top}\boldsymbol{a}, b + \boldsymbol{t}^\top L^{-\top}\boldsymbol{a}), \quad g = (L, \boldsymbol{t}) \in GL(m) \ltimes \mathbb{R}^m, \ (\boldsymbol{a}, b) \in \Xi. \tag{21}$$

Then, we can see the feature map $\phi(\boldsymbol{x}, (\boldsymbol{a}, b)) := \sigma(\boldsymbol{a} \cdot \boldsymbol{x} - b)$ is joint-$G$-invariant. In fact,

$$\phi(g \cdot \boldsymbol{x}, g \cdot (\boldsymbol{a}, b)) = \sigma\left(L^{-\top}\boldsymbol{a} \cdot (L\boldsymbol{x} + \boldsymbol{t}) - (b + \boldsymbol{t}^\top L^{-\top}\boldsymbol{a})\right) = \sigma(\boldsymbol{a} \cdot \boldsymbol{x} - b) = \phi(\boldsymbol{x}, (\boldsymbol{a}, b)).$$

By Lemma 3, the regular representation $\pi$ of $G = \mathrm{Aff}(m)$ is irreducible. Therefore, by Theorem 3, the depth-2 neural network and corresponding ridgelet transform:

$$\mathrm{NN}[\gamma](\boldsymbol{x}) = \int_{\mathbb{R}^m \times \mathbb{R}} \gamma(\boldsymbol{a}, b)\sigma(\boldsymbol{a} \cdot \boldsymbol{x} - b)\mathrm{d}\boldsymbol{a}\mathrm{d}b, \quad \text{and} \quad \mathrm{R}_2[f](\boldsymbol{a}, b) = \int_{\mathbb{R}^m} f(\boldsymbol{x})\overline{\rho(\boldsymbol{a} \cdot \boldsymbol{x} - b)}\mathrm{d}\boldsymbol{x},$$

satisfy the reconstruction formula $\mathrm{NN} \circ \mathrm{R}_2 = ((\sigma, \rho)) \, \mathrm{Id}_{L^2(\mathbb{R}^m)}$. In Appendix A, we supplemented a detailed proof. In Appendix B, we discussed a geometric interpretation of dual $G$-action (21).

## 4.2 Depth-$n$

Following Corollary 1, we derive the ridgelet transform for depth-$n$ fully-connected network by constructing a joint-equivariant network.

Let $O(m)$ be the $m$-dimensional orthogonal group acting on $\mathbb{R}^m$ by $Q\boldsymbol{v}$ for $Q \in O(m)$ and $\boldsymbol{v} \in \mathbb{R}^m$, and (re)set $G := O(m) \times \mathrm{Aff}(m)$ be the product group, acting on the data domain $X$ by

$$g \cdot \boldsymbol{x} := L\boldsymbol{x} + \boldsymbol{t}, \quad \boldsymbol{x} \in X, g = (Q, L, \boldsymbol{t}) \in G = O(m) \times (GL(m) \ltimes \mathbb{R}^m). \tag{22}$$

Namely, we set the $O(m)$-action on $X$ is trivial. Define a unitary representation $\pi$ of $G$ on vector-valued square-integrable functions $L^2(X; X)$ as

$$\pi_g[\boldsymbol{f}](\boldsymbol{x}) := Q\boldsymbol{f}(L^{-1}(\boldsymbol{x} - \boldsymbol{t})), \quad \boldsymbol{x} \in X, g = (Q, L, \boldsymbol{t}) \in G, \boldsymbol{f} \in L^2(X; X). \tag{23}$$

**Lemma 5.** *The above $\pi : G \to L^2(\mathbb{R}^m; \mathbb{R}^m)$ is an irreducible unitary representation.*

*Proof.* Recall that a tensor product of irreducible representations is irreducible. Since both $O(m)$-action on $\mathbb{R}^m$ and $\mathrm{Aff}(m)$-action on $L^2(\mathbb{R}^m)$ are irreducible, and $L^2(\mathbb{R}^m; \mathbb{R}^m)$ is a tensor product $\mathbb{R}^m \otimes L^2(\mathbb{R}^m)$, so the action $\pi$ of product group $O(m) \times \mathrm{Aff}(m)$ on tensor product $\mathbb{R}^m \otimes L^2(\mathbb{R}^m) = L^2(\mathbb{R}^m; \mathbb{R}^m)$ is irreducible. $\qquad\square$

Following the same arguments in Lemma 1, we first construct a *depth*-2 joint-$G$-equivariant network. Take an arbitrary square-integrable (not yet joint-$G$-equivariant) vector-field $\boldsymbol{f}_0 \in L^2(X; X)$. Then, the following network is joint-$G$-equivariant:

$$\mathrm{NN}(\boldsymbol{x}, \xi) := \mathrm{NN}[\mathrm{R}_2[\pi_\xi[\boldsymbol{f}_0]]](\boldsymbol{x}) = \int_{\mathbb{R}^m \times \mathbb{R}} Q\mathrm{R}_2[\boldsymbol{f}_0](\boldsymbol{a}, b)\sigma\left(\boldsymbol{a}^\top L^{-1}(\boldsymbol{x} - \boldsymbol{t}) - b\right) \mathrm{d}\boldsymbol{a}\mathrm{d}b, \tag{24}$$

for every $\boldsymbol{x} \in X, \xi = (Q, L, \boldsymbol{t}) \in O(m) \times GL(m) \ltimes \mathbb{R}^m$. Here $\mathrm{R}_2$ is the ridgelet transform for depth-2 case (applied for vector-valued function by element-wise manner). This is joint-$G$-equivariant because $\mathrm{NN}(\boldsymbol{x}, \xi) = \pi_\xi[\boldsymbol{f}_0](\boldsymbol{x})$. Henceforth, we (re)set $\Xi := G$.

Finally, we construct a *depth-$n$* joint-$G$-equivariant network by composing the above depth-2 networks as below. Write $\boldsymbol{\xi} := (\xi_1, \ldots, \xi_n) \in \Xi^n$ for short. For any measurable function $\gamma : \Xi^n \to \mathbb{C}$ and vector-field $\boldsymbol{f} : \mathbb{R}^m \to \mathbb{R}^m$, put

$$\mathrm{DNN}(\boldsymbol{x}) := \int_{\Xi^n} \gamma(\boldsymbol{\xi})\mathrm{NN}(\bullet, \xi_n) \circ \cdots \circ \mathrm{NN}(\boldsymbol{x}, \xi_1)\mathrm{d}\boldsymbol{\xi}, \quad \boldsymbol{x} \in X \tag{25}$$

$$\mathrm{R}_n[\boldsymbol{f}](\boldsymbol{\xi}) := \int_X \boldsymbol{f}(\boldsymbol{x})^\top \mathrm{NN}(\bullet, \xi_n) \circ \cdots \circ \mathrm{NN}(\boldsymbol{x}, \xi_1)\mathrm{d}\boldsymbol{x}, \quad \boldsymbol{\xi} \in \Xi^n. \tag{26}$$

Then, as a consequence of Corollary 1, there exists a constant $c \in \mathbb{C}$ satisfying $\mathrm{DNN} \circ \mathrm{R}_n[\boldsymbol{f}] = c\boldsymbol{f}$ for any $\boldsymbol{f} \in L^2(X; X)$.

## 5  Example: Formal Deep Network

We explain the *formal deep network* (FDN) introduced by Sonoda et al. [32]. Compared to the depth-$n$ fully-connected network introduced in the previous section, the FDN (introduced in the previous study) is more abstract because the network architecture is not specified. Yet, we consider this is still useful for theoretical study of deep networks as it covers a wide range of groups and data domains (i.e., not limited to the affine group and the Euclidean space).

### 5.1  Formal Deep Network

Let $G$ be an arbitrary locally compact group equipped with left-invariant measure $\mathrm{d}g$, let $X$ be a $G$-space equipped with left-invariant measure $\mathrm{d}x$, and set $\Xi := G$ with right-invariant measure $\mathrm{d}\xi$. The key concept is to identify each feature map $\phi : X \times \Xi \to X$ with a $G$-action $g : X \to X$ with parameter domain $\Xi$ being identified with group $G$, and the composite of feature maps, say $g \circ h$, with product $gh$. Since a group is closed under its operation by definition, the proposed network can represent literally *any depth* such as a single hidden layer $g$, double hidden layers $g \circ h$, triple hidden layers $g \circ h \circ k$, and infinite hidden layers $g \circ h \circ \cdots$. Besides, to lift the group action on a linear space, the network is formulated as a regular action of group $G$ on a hidden layer, say $\psi \in L^2(X)$.

**Definition 6** (Formal Deep Network). For any functions $\psi \in L^2(X)$ and $\gamma : \Xi \to \mathbb{C}$, put

$$\mathtt{DNN}[\gamma; \psi](x) := \int_{G_1 \rtimes \cdots \rtimes G_n} \gamma(\xi_1, \ldots, \xi_n)\, \psi \circ \xi_n \circ \cdots \circ \xi_1(x) \mathrm{d}\xi_1 \cdots \mathrm{d}\xi_n, \quad x \in X. \qquad (27)$$

Here, $G = G_1 \rtimes \cdots \rtimes G_n$ denotes the semi-direct product of groups, suggesting that the network gets much complex and expressive as it gets deeper.

To see the universality, define the dual action of $G$ on the parameter domain $\Xi = G$ as

$$g \cdot \xi := \xi g^{-1}, \quad g \in G, \xi \in \Xi. \qquad (28)$$

Then, we can see $\phi(x, \xi) := \psi \circ \xi(x)$ is joint-$G$-*invariant*. In fact,

$$\phi(g \cdot x, g \cdot \xi) = \psi \circ (g \cdot \xi)(g \cdot x) = \psi \circ (\xi \circ g^{-1})(g(x)) = \psi \circ \xi(x) = \phi(x, \xi).$$

Therefore, by Theorem 3, assuming that the regular representation $\pi : G \to \mathcal{U}(L^2(X))$ is irreducible, the ridgelet transform is given by

$$\mathtt{R}[f](\xi_1, \ldots, \xi_n) = \int_X f(x)\overline{\psi \circ \xi_n \circ \cdots \circ \xi_1(x)}\mathrm{d}x, \quad (\xi_1, \ldots, \xi_n) \in G_1 \rtimes \cdots \rtimes G_n \qquad (29)$$

satisfying $\mathtt{NN} \circ \mathtt{R} = (\!(\sigma, \rho)\!)\, \mathrm{Id}_{L^2(X)}$.

### 5.2  Depth Separation

To enjoy the advantage of abstract formulation, we discuss the effect of depth. For the sake of simplicity, we assume $G$ to be a finite group, which may be acceptable given that the data domain $X$ in practice is often discretized (or coarse-grained) into finite sets of representative points, say $X \approx \overline{X} := \{x_i\}_{i=1}^p$, and if so the $G$-action is also reduced to finite representative actions.

Following the concept of the formal deep network, we call group $G$ acting on $X$ a network. Let us consider depth-1 network $G$ and depth-$n$ network $G_1 \rtimes \cdots \rtimes G_n$ satisfying $G = G_1 \rtimes \cdots \rtimes G_n$. The equation indicates that two networks have the same expressive power, because they can implement the same class of maps $g : X \to X$.

Next, let us define the *width* of a single layer $G$ as the cardinality $|G|$. This is reasonable because the set $G$ parametrizes each map $g : X \to X$. Then, under the assumption that each $G_i$ is simple, the depth-$n$ network $G_1 \rtimes \cdots \rtimes G_n$ can express the same class of depth-1 network exponentially-effectively, because the total widths are $\sum_{i=1}^n |G_i| = O(n)$ for depth-$n$ and $\prod_{i=1}^n |G_i| = \exp O(n)$ for depth-1. This estimate can be interpreted as the classical thought that the hierarchical models such as deep networks can represent complex functions combinatorially more efficient than shallow models.

## 6 Discussion

We have developed a systematic method for deriving a ridgelet transform for a wide range of learning machines defined by joint-group-equivariant feature maps, yielding the universal approximation theorems as corollaries. The previous results by Sonoda et al. [33] was limited to scalar-valued joint-invariant functions, which were insufficient to deal with practical learning machines defined by composite mappings of vector-valued functions, such as deep neural networks. For example, they could only deal with abstract composite structures like formal deep network [32]. By extending their argument to vector-valued joint-equivariant functions, we were able to deal with deep structures. Traditionally, the techniques used in the expressive power analysis of deep networks were different from those used in the analysis of shallow networks, as overviewed in the introduction. Nonetheless, our main theorem cover both deep and shallow networks from the unified perspective (joint-group-action on the data-parameter domain). Technically, this unification is due to Schur's lemma, a basic and useful result in the representation theory. Thanks to this lemma, the proof of the main theorem is simple, yet the scope of application is wide. The significance of this study lies in revealing the close relationship between machine learning theory and modern algebra. With this study as a catalyst, we expect a major upgrade to machine learning theory from the perspective of modern algebra.

### 6.1 Limitations

In the main theorem, we assume the following: (1) joint-equivariance of feature map $\phi$, (2) boundedness of composite operator $\mathtt{NN} \circ \mathtt{R}$, (3) irreducibility of unitary representation $\pi$. In addition, throughout this study, we assume (4) local compactness of group $G$, and (5) that the network is given by the integral representation.

As discussed in the main text, satisfying (1) is much easier than (non-joint) equivariance. Also, (2) is often a textbook excersise when the specific expression is given. (3) is required for Schur's lemma, and it is often sufficient to synthesize the known results such as the one for the example of depth-$n$ fully-connected network. (4) is quite a frequent assumption in the standard group representation theory, but it excludes infinite-dimensional groups. When formulated *natively*, nonparametric learning models including DNN can be infinite-dimensional groups. However, from the perspective of learnability, it is nonsense to consider too large a model, and it is common to assume regularity conditions such as sparsity and low rank in usual theoretical analysis. So, it is natural to impose additional regularity conditions for satisfying local compactness. (5) may be rather an advantage because there are established techniques to show the $cc$-universaity of finite models by discretizing integral representations. Moreover, there is a fast discretization scheme called the Barron's rate based on the quasi-Monte Carlo method. On the other hand, problems like the minimum width in the field of deep narrow networks are analyses of finite parameters, and they could be a different type of parameters. Yet, the current mainstream solutions are the information theoretic method by Park et al. [23] and the neural ODE method by Cai [2], and both arguments contain the discretization of continuous models. Therefore, we may expect a high affinity with the integral representation theory.

This study is the first step in extending the harmonic analysis method, which was previously applicable only to shallow models, to deep models. The above limitations will be resolved in our future works.

## 7 Broader Impact

This work studies theoretical aspects of neural networks for expressing square integrable functions. Since we do not propose a new method nor a new dataset, we expect that the impact of this work on ethical aspects and future societal consequences will be small, if any. Our work can help understand the theoretical benefit and limitations of neural networks in approximating functions. Our work and the proof technique improve our understanding of the theoretical aspect of deep neural networks and other learning machines used in machine learning, and may lead to better use of these techniques with possible benefits to the society.

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

## A  Depth-2 Fully-Connected Neural Network and Ridgelet Transform

A non group theoretic proof by reducing to a Fourier expression is given in Sonoda et al. [29, Theorem 6].

## A.1 Proof

In the following, we identify the group $G$ acting on data domain $\mathbb{R}^m$ with the affine group $\mathrm{Aff}(\mathbb{R}^m)$, and introduce the so-called twisted dual group action that leaves a function $\theta$ invariant. Then, we see the regular action $\pi$ of $G$ on functions space $L^2(\mathbb{R}^m)$ commutes with composite $S_\sigma \circ R_\rho$. Hence, by Schur's lemma, $S_\sigma \circ R_\rho$ is a constant multiple of identity, which concludes the assertion.

*Proof.* Let $G$ be the affine group $\mathrm{Aff}(\mathbb{R}^m) = GL(\mathbb{R}^m) \ltimes \mathbb{R}^m$. For any $g = (L, \boldsymbol{t}) \in G$, let

$$g \cdot \boldsymbol{x} := L\boldsymbol{x} + \boldsymbol{t}, \quad \boldsymbol{x} \in \mathbb{R}^m \tag{30}$$

be its action on $\mathbb{R}^m$, and let

$$\begin{aligned}\pi(g)[f](\boldsymbol{x}) &:= |\det L|^{-1/2} f(g^{-1} \cdot \boldsymbol{x}) \\ &= |\det L|^{-1/2} f(L^{-1}(\boldsymbol{x} - \boldsymbol{t})), \quad f \in L^2(\mathbb{R}^m)\end{aligned} \tag{31}$$

be its left-regular action on $L^2(\mathbb{R}^m)$.

Besides, putting

$$\theta((\boldsymbol{a}, b), \boldsymbol{x}) := \boldsymbol{a} \cdot \boldsymbol{x} - b, \quad (\boldsymbol{a}, b) \in \mathbb{R}^m \times \mathbb{R}, \boldsymbol{x} \in \mathbb{R}^m \tag{32}$$

we define the *twisted dual action* of $G$ on $\mathbb{R}^m \times \mathbb{R}$ as

$$g \cdot (\boldsymbol{a}, b) := (L^{-\top}\boldsymbol{a}, b + \boldsymbol{a} \cdot (L^{-1}\boldsymbol{t})), \quad (\boldsymbol{a}, b) \in \mathbb{R}^m \times \mathbb{R} \tag{33}$$

so that the following invariance hold:

$$\theta(g \cdot (\boldsymbol{a}, b), g \cdot \boldsymbol{x}) = \theta((\boldsymbol{a}, b), \boldsymbol{x}) = \boldsymbol{a} \cdot \boldsymbol{x} - b. \tag{34}$$

To see this, use matrix expressions with extended variables

$$\theta((\boldsymbol{a}, b), \boldsymbol{x}) = \begin{pmatrix} \boldsymbol{a}^\top & b \end{pmatrix} \begin{pmatrix} I_m & 0 \\ 0 & -1 \end{pmatrix} \begin{pmatrix} \boldsymbol{x} \\ 1 \end{pmatrix} =: \tilde{\boldsymbol{a}}^\top \tilde{I} \tilde{\boldsymbol{x}}, \tag{35}$$

$$\widetilde{g \cdot \boldsymbol{x}} := \begin{pmatrix} g \cdot \boldsymbol{x} \\ 1 \end{pmatrix} = \begin{pmatrix} L & \boldsymbol{t} \\ 0 & 1 \end{pmatrix} \begin{pmatrix} \boldsymbol{x} \\ 1 \end{pmatrix} =: \tilde{L}\tilde{\boldsymbol{x}} \tag{36}$$

and calculate

$$\tilde{\boldsymbol{a}}^\top \tilde{I} \tilde{\boldsymbol{x}} = (\tilde{\boldsymbol{a}}^\top \tilde{I} \tilde{L}^{-1} \tilde{I}^{-1}) \tilde{I}(\tilde{L}\tilde{\boldsymbol{x}}) = (\tilde{I}\tilde{L}^{-\top}\tilde{I}\tilde{\boldsymbol{a}})^\top \tilde{I}(\tilde{L}\tilde{\boldsymbol{x}}), \tag{37}$$

which suggests $\widetilde{g \cdot (\boldsymbol{a}, b)} := \tilde{I}\tilde{L}^{-\top}\tilde{I}\tilde{\boldsymbol{a}}$, and we have

$$\begin{aligned}\tilde{I}\tilde{L}^{-\top}\tilde{I} &= \begin{pmatrix} I_m & 0 \\ 0 & -1 \end{pmatrix} \begin{pmatrix} L & \boldsymbol{t} \\ 0 & 1 \end{pmatrix}^{-\top} \begin{pmatrix} I_m & 0 \\ 0 & -1 \end{pmatrix} \\ &= \begin{pmatrix} I_m & 0 \\ 0 & -1 \end{pmatrix} \begin{pmatrix} L^{-\top} & 0 \\ -\boldsymbol{t}^\top L^{-\top} & 1 \end{pmatrix} \begin{pmatrix} I_m & 0 \\ 0 & -1 \end{pmatrix} = \begin{pmatrix} L^{-\top} & 0 \\ \boldsymbol{t}^\top L^{-\top} & 1 \end{pmatrix}.\end{aligned}$$

Further, we define its regular-action by

$$\begin{aligned}\widehat{\pi}(g)[\gamma](\boldsymbol{a}, b) &:= |\det L|^{1/2} \gamma(g^{-1} \cdot (\boldsymbol{a}, b)) \\ &= |\det L|^{1/2} \gamma(L^\top \boldsymbol{a}, b - \boldsymbol{a} \cdot \boldsymbol{t}), \quad (\boldsymbol{a}, b) \in \mathbb{R}^m \times \mathbb{R}.\end{aligned} \tag{38}$$

Then we can see that, for all $g = (L, \boldsymbol{t}) \in G$,

$$R_\rho \circ \pi(g) = \widehat{\pi}(g) \circ R_\rho, \quad \text{and} \quad S_\sigma \circ \widehat{\pi}(g) = \pi(g) \circ S_\sigma. \tag{39}$$

In fact, at every $g = (L, \boldsymbol{t}) \in G$ and $(\boldsymbol{a}, b) \in \mathbb{R}^m \times \mathbb{R}$,

$$R_\rho[\pi(g)[f]](\boldsymbol{a}, b) = |\det L|^{-1/2} \int_{\mathbb{R}^m} f(g^{-1} \cdot \boldsymbol{x}) \overline{\rho(\theta((\boldsymbol{a}, b), \boldsymbol{x}))} \mathrm{d}\boldsymbol{x}$$

by putting $\boldsymbol{x} = g \cdot \boldsymbol{y} = L\boldsymbol{y} + \boldsymbol{t}$ with $\mathrm{d}\boldsymbol{x} = |\det L|\mathrm{d}\boldsymbol{y}$,

$$= |\det L|^{1/2} \int_{\mathbb{R}^m} f(\boldsymbol{y}) \overline{\rho(\theta((\boldsymbol{a}, b), g \cdot \boldsymbol{y}))} \mathrm{d}\boldsymbol{y}$$

$$= |\det L|^{1/2} \int_{\mathbb{R}^m} f(\boldsymbol{y}) \overline{\rho(\theta(\theta(g^{-1} \cdot (\boldsymbol{a}, b)), \boldsymbol{y})))} \mathrm{d}\boldsymbol{y}$$

$$= \widehat{\pi}(g)[R_\rho[f]](\boldsymbol{a}, b). \tag{40}$$

Similarly, at every $g = (L, \boldsymbol{t}) \in G$ and $\boldsymbol{x} \in \mathbb{R}^m$,

$$S_\sigma[\widehat{\pi}(g)[\gamma]](\boldsymbol{x}) = |\det L|^{1/2} \int_{\mathbb{R}^m \times \mathbb{R}} \gamma(g^{-1} \cdot (\boldsymbol{a}, b)) \sigma(\theta((\boldsymbol{a}, b), \boldsymbol{x})) \mathrm{d}\boldsymbol{a}\mathrm{d}b$$

by putting $(\boldsymbol{a}, b) := g \cdot (\boldsymbol{\xi}, \eta) = (L^{-\top}\boldsymbol{\xi}, \eta + \boldsymbol{\xi} \cdot (L^{-1}\boldsymbol{t}))$ with $\mathrm{d}\boldsymbol{a}\mathrm{d}b = |\det L|\mathrm{d}\boldsymbol{\xi}\mathrm{d}\eta$,

$$= |\det L|^{-1/2} \int_{\mathbb{R}^m \times \mathbb{R}} \gamma(\boldsymbol{\xi}, \eta) \sigma(\theta(g \cdot (\boldsymbol{\xi}, \eta), \boldsymbol{x})) \mathrm{d}\boldsymbol{\xi}\mathrm{d}\eta$$

$$= |\det L|^{-1/2} \int_{\mathbb{R}^m \times \mathbb{R}} \gamma(\boldsymbol{\xi}, \eta) \sigma(\theta((\boldsymbol{\xi}, \eta), g^{-1} \cdot \boldsymbol{x})) \mathrm{d}\boldsymbol{\xi}\mathrm{d}\eta$$

$$= \pi(g)[S_\sigma[\gamma]](\boldsymbol{x}). \tag{41}$$

Hence $S_\sigma \circ R_\rho$ commutes with $\pi(g)$ because

$$S_\sigma \circ R_\rho \circ \pi(g) = S_\sigma \circ \widehat{\pi}(g) \circ R_\rho = \pi(g) \circ S_\sigma \circ R_\rho.$$

Since $S_\sigma \circ R_\rho : L^2(\mathbb{R}^m) \to L^2(\mathbb{R}^m)$ is bounded (Lemma 4), and $(\pi, L^2(\mathbb{R}^m))$ is an irreducible unitary representation of $G$ (Lemma 3), Schur's lemma (Theorem 2) yields that there exist a constant $C_{\sigma,\rho} \in \mathbb{C}$ such that

$$S_\sigma \circ R_\rho[f] = C_{\sigma,\rho} f \tag{42}$$

for all $f \in L^2(\mathbb{R}^m)$.

Finally, by directly computing the left-hand-side, namely $S_\sigma \circ R_\rho[f]$, we can verify that the constant $C_{\sigma,\rho}$ is given by

$$C_{\sigma,\rho} = ((\sigma, \rho)) := (2\pi)^{m-1} \int_{\mathbb{R}} \sigma^\sharp(\omega) \overline{\rho^\sharp(\omega)} |\omega|^{-m} \mathrm{d}\omega. \tag{43}$$

$\square$

## A.2 Proof for (33)

Use matrix expressions with extended variables

$$\theta((\boldsymbol{a}, b), \boldsymbol{x}) = \begin{pmatrix} \boldsymbol{a}^\top & b \end{pmatrix} \begin{pmatrix} I_m & 0 \\ 0 & -1 \end{pmatrix} \begin{pmatrix} \boldsymbol{x} \\ 1 \end{pmatrix} =: \tilde{\boldsymbol{a}}^\top \tilde{I} \tilde{\boldsymbol{x}}, \tag{44}$$

$$\widetilde{g \cdot \boldsymbol{x}} := \begin{pmatrix} g \cdot \boldsymbol{x} \\ 1 \end{pmatrix} = \begin{pmatrix} L & \boldsymbol{t} \\ 0 & 1 \end{pmatrix} \begin{pmatrix} \boldsymbol{x} \\ 1 \end{pmatrix} =: \tilde{L} \tilde{\boldsymbol{x}} \tag{45}$$

and calculate

$$\tilde{\boldsymbol{a}}^\top \tilde{I} \tilde{\boldsymbol{x}} = (\tilde{\boldsymbol{a}}^\top \tilde{I} \tilde{L}^{-1} \tilde{I}^{-1}) \tilde{I}(\tilde{L}\tilde{\boldsymbol{x}}) = (\tilde{I}\tilde{L}^{-\top}\tilde{I}\tilde{\boldsymbol{a}})^\top \tilde{I}(\tilde{L}\tilde{\boldsymbol{x}}), \tag{46}$$

which suggests $\widetilde{g \cdot (\boldsymbol{a}, b)} := \tilde{I}\tilde{L}^{-\top}\tilde{I}\tilde{\boldsymbol{a}}$, and we have

$$\tilde{I}\tilde{L}^{-\top}\tilde{I} = \begin{pmatrix} I_m & 0 \\ 0 & -1 \end{pmatrix} \begin{pmatrix} L & \boldsymbol{t} \\ 0 & 1 \end{pmatrix}^{-\top} \begin{pmatrix} I_m & 0 \\ 0 & -1 \end{pmatrix}$$

$$= \begin{pmatrix} I_m & 0 \\ 0 & -1 \end{pmatrix} \begin{pmatrix} L^{-\top} & 0 \\ -\boldsymbol{t}^\top L^{-\top} & 1 \end{pmatrix} \begin{pmatrix} I_m & 0 \\ 0 & -1 \end{pmatrix} = \begin{pmatrix} L^{-\top} & 0 \\ \boldsymbol{t}^\top L^{-\top} & 1 \end{pmatrix}.$$

 **A.3 Proof for** (39)

456   In fact, at every $g = (L, \boldsymbol{t}) \in G$ and $(\boldsymbol{a}, b) \in \mathbb{R}^m \times \mathbb{R}$,

$$R_\rho[\pi(g)[f]](\boldsymbol{a}, b) = |\det L|^{-1/2} \int_{\mathbb{R}^m} f(g^{-1} \cdot \boldsymbol{x}) \overline{\rho(\theta((\boldsymbol{a}, b), \boldsymbol{x}))} \mathrm{d}\boldsymbol{x}$$

457   by putting $\boldsymbol{x} = g \cdot \boldsymbol{y} = L\boldsymbol{y} + \boldsymbol{t}$ with $\mathrm{d}\boldsymbol{x} = |\det L| \mathrm{d}\boldsymbol{y}$,

$$= |\det L|^{1/2} \int_{\mathbb{R}^m} f(\boldsymbol{y}) \overline{\rho(\theta((\boldsymbol{a}, b), g \cdot \boldsymbol{y})))} \mathrm{d}\boldsymbol{y}$$

$$= |\det L|^{1/2} \int_{\mathbb{R}^m} f(\boldsymbol{y}) \overline{\rho(\theta(g^{-1} \cdot (\boldsymbol{a}, b), \boldsymbol{y})))} \mathrm{d}\boldsymbol{y}$$

$$= \widehat{\pi}(g)[R_\rho[f]](\boldsymbol{a}, b). \tag{47}$$

458   Similarly, at every $g = (L, \boldsymbol{t}) \in G$ and $\boldsymbol{x} \in \mathbb{R}^m$,

$$S_\sigma[\widehat{\pi}(g)[\gamma]](\boldsymbol{x}) = |\det L|^{1/2} \int_{\mathbb{R}^m \times \mathbb{R}} \gamma(g^{-1} \cdot (\boldsymbol{a}, b)) \sigma(\theta((\boldsymbol{a}, b), \boldsymbol{x})) \mathrm{d}\boldsymbol{a} \mathrm{d}b$$

459   by putting $(\boldsymbol{a}, b) := g \cdot (\boldsymbol{\xi}, \eta) = (L^{-\top}\boldsymbol{\xi}, \eta + \boldsymbol{\xi} \cdot (L^{-1}\boldsymbol{t}))$ with $\mathrm{d}\boldsymbol{a}\mathrm{d}b = |\det L| \mathrm{d}\boldsymbol{\xi}\mathrm{d}\eta$,

$$= |\det L|^{-1/2} \int_{\mathbb{R}^m \times \mathbb{R}} \gamma(\boldsymbol{\xi}, \eta) \sigma(\theta(g \cdot (\boldsymbol{\xi}, \eta), \boldsymbol{x})) \mathrm{d}\boldsymbol{\xi}\mathrm{d}\eta$$

$$= |\det L|^{-1/2} \int_{\mathbb{R}^m \times \mathbb{R}} \gamma(\boldsymbol{\xi}, \eta) \sigma(\theta((\boldsymbol{\xi}, \eta), g^{-1} \cdot \boldsymbol{x})) \mathrm{d}\boldsymbol{\xi}\mathrm{d}\eta$$

$$= \pi(g)[S_\sigma[\gamma]](\boldsymbol{x}). \tag{48}$$

460

# B   Geometric Interpretation of Dual Action for Original Ridgelet Transform

462   We explain a geometric interpretation of the dual action (33) in the previous section. We note that
463   in general $\theta$ does not require any geometric interpretation as long as it is joint group invariant on
464   data-parameter domain.

465   For each $(\boldsymbol{a}, b) \in \mathbb{R}^m \times \mathbb{R}$, put $\xi(\boldsymbol{a}, b) := \{\boldsymbol{x} \in \mathbb{R}^m \mid \boldsymbol{a} \cdot \boldsymbol{x} - b = 0\}$. Then it is a hyperplane in $\mathbb{R}^m$
466   through point $\boldsymbol{x}_0 = b\boldsymbol{a}/|\boldsymbol{a}|^2$ with normal vector $\boldsymbol{u} := \boldsymbol{a}/|\boldsymbol{a}|$.

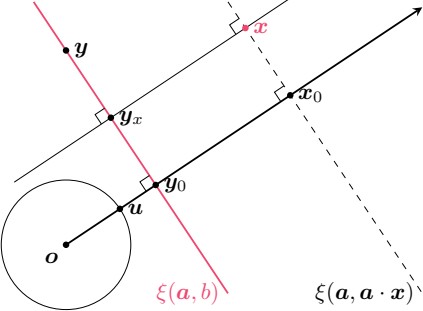

Figure 3: The invariant $\phi((\boldsymbol{a}, b), \boldsymbol{x}) = \sigma(\boldsymbol{a} \cdot \boldsymbol{x} - b)$ is the euclidean distance between point $\boldsymbol{x}$ and hyperplane $\xi(\boldsymbol{a}, b)$ followed by scaling and nonlinearity $\sigma$

467   For any point $\boldsymbol{y}$ in the hyperplane $\xi(\boldsymbol{a}, b)$, by definition $\boldsymbol{a} \cdot \boldsymbol{y} = b$, thus

$$\boldsymbol{a} \cdot \boldsymbol{x} - b = \boldsymbol{a} \cdot (\boldsymbol{x} - \boldsymbol{y}). \tag{49}$$

468   But this means $\boldsymbol{a} \cdot \boldsymbol{x} - b$ is a scaled distance between point $\boldsymbol{x}$ and hyperplane $\xi(\boldsymbol{a}, b)$,

$$= |\boldsymbol{a}| d_E(\boldsymbol{x}, \xi(\boldsymbol{a}, b)), \tag{50}$$

and further a scaled distance between hyperplanes $\xi(\boldsymbol{a}, \boldsymbol{a} \cdot \boldsymbol{x})$ through $\boldsymbol{x}$ with normal $\boldsymbol{a}/|\boldsymbol{a}|$ and $\xi(\boldsymbol{a}, b)$,

$$= |\boldsymbol{a}| d_E(\xi(\boldsymbol{a}, \boldsymbol{a} \cdot \boldsymbol{x}), \xi(\boldsymbol{a}, b)). \tag{51}$$

Now, we can interpret the invariant $\theta((\boldsymbol{a}, b), \boldsymbol{x}) := \boldsymbol{a} \cdot \boldsymbol{x} - b$ in a geometric manner, that is, it is the distance between point and hyperplane, or between hyperplanes. We note that we can regard entire $\sigma(\boldsymbol{a} \cdot \boldsymbol{x} - b)$—the distance modulated by both scaling and nonlinearity—as the invariant, say $\phi$.

Furthermore, the dual action $g \cdot (\boldsymbol{a}, b)$ is understood as a parallel translation of hyperplane $\xi(\boldsymbol{a}, b)$ to $\xi(g \cdot (\boldsymbol{a}, b))$ so as to leave the scaled distance $\theta$ invariant, namely

$$d_E(g \cdot \boldsymbol{x}, g \cdot \xi(\boldsymbol{a}, b)) = d_E(\boldsymbol{x}, \xi(\boldsymbol{a}, b)). \tag{52}$$

Indeed, for any $g = (L, \boldsymbol{t}) \in G$,

$$\begin{aligned}
g \cdot \xi(\boldsymbol{a}, b) &= \{g \cdot \boldsymbol{x} \mid \boldsymbol{a} \cdot \boldsymbol{x} - b = 0\} \\
&= \{\boldsymbol{y} \mid \boldsymbol{a} \cdot (g^{-1} \cdot \boldsymbol{y}) - b = 0\} && \text{(by letting } \boldsymbol{y} = g \cdot \boldsymbol{x}) \\
&= \{\boldsymbol{y} \mid (L^{-\top}) \cdot \boldsymbol{y} - (b + \boldsymbol{a} \cdot (L^{-1}\boldsymbol{t})) = 0\} \\
&= \xi(g \cdot (\boldsymbol{a}, b)),
\end{aligned}$$

meaning that the hyperplane with parameter $(\boldsymbol{a}, b)$ translated by $g$ is identical to the hyperplane with parameter $g \cdot (\boldsymbol{a}, b)$.

To summarize, in the case of fully-connected neural network (and its corresponding ridgelet transform), the invariant is a modulated distance $\sigma(\boldsymbol{a} \cdot \boldsymbol{x} - b)$, and the dual action is the parallel translation of hyperplane so as to keep the distance invariant. Further, from this geometric perspective, we can rewrite the fully-connected neural network in a geometric manner as

$$S[\gamma](\boldsymbol{x}) := \int_{\mathbb{R} \times \Xi} \gamma(\xi) \sigma(a d_E(\boldsymbol{x}, \xi)) \mathrm{d}a \mathrm{d}\xi, \tag{53}$$

where $a \in \mathbb{R}$ denotes signed scale and $\Xi$ denotes the space of all hyperplanes (not always through the origin). Since each hyperplane is parametrized by normal vectors $\boldsymbol{u} \in {}^{m-1}$ and distance $p \geq 0$ from the origin, we can induce the product of spherical measure $\mathrm{d}\boldsymbol{u}$ and Lebesgue measure $\mathrm{d}p$ as a measure $\mathrm{d}\xi$ on the space $\Xi$ of hyperplanes.

