# OpenReview forum: "Constructive Universal Approximation Theorems for Deep Joint-Equivariant Networks by Schur's Lemma"
_NeurIPS.cc/2024/Conference — Submitted to NeurIPS 2024_

### Official Review · Reviewer_bUuw · 2024-06-22

**Soundness:** 3
**Presentation:** 3
**Contribution:** 2
**Rating:** 5
**Confidence:** 3

**Summary:**

The paper presents a unified approach to universal approximation theorems for neural networks using group representation theory. It extends to vector-valued joint-group-equivariant feature maps, providing a systematic method for both shallow and deep neural networks with nonlinear activation functions. By leveraging Schur's lemma, the paper shows that these networks can universally approximate any function within a certain class. It main contribution is the closed-form ridgelet transform, which offers a constructive proof and explicit parameter distribution for these networks.

**Strengths:**

1.The paper introduces a unified constructive universal approximation theorem that applies to both shallow and deep neural networks using group representation theory. This is an innovative approach. It also extends previous work by incorporating vector-valued joint-group-equivariant feature maps.

2.  The paper is theoretically sounding, leveraging concepts from group representation theory and Schur's lemma. They perform the thorough and systematic development of the ridgelet transform, providing a closed-form solution for parameter distributions and ensuring the findings are theoretically well justified.

3. The paper is well-structured and clearly written. Definitions, theorems, and proofs are presented in a coherent manner, making it easier for readers to follow the details of the argument and understand the implications of the results.

4.  This work is significant since it provides a relationship between deep learning theory and modern algebra. By providing a unified framework that applies to a wide range of network architectures, the paper incentivize further research and development in the field of machine learning.

**Weaknesses:**

1.  While the paper is strong in its theoretical contributions, it lacks empirical validation through experiments or simulations. Demonstrating the practical applicability and effectiveness of the proposed ridgelet transform and the unified framework on real-world datasets or benchmark problems would strengthen the paper. Including even a small set of experiments could provide evidence of the practical relevance and performance of the theoretical results.

2. This work makes several assumptions, such as the local compactness of the group \( G \) and the boundedness of the composite operator \( \text{NN} \circ R \). While these assumptions are standard in group representation theory, the paper could benefit from a more detailed discussion on their implications and limitations. Exploring scenarios where these assumptions might not hold or providing guidance on how to relax these assumptions.

3. Some of the technical details, particularly those related to advanced concepts in group representation theory and the ridgelet transform, might be challenging for readers who are not experts in these areas. Providing additional intuitive explanations, diagrams, or examples to illustrate these concepts could enhance the clarity of the paper.

**Questions:**

1. The work introduces a theoretically robust framework for universal approximation using the ridgelet transform and group representation theory. How feasible is it to implement these theoretical constructs in practical neural network architectures? Have the authors considered the computational complexity and resource requirements for applying these methods to real-world datasets, and if so, what strategies or approximations can be employed to make this approach computationally efficient?

2. The main results rely on several key assumptions, such as the local compactness of the group \( G \) and the irreducibility of the unitary representation \( \pi \). How robust are the findings to deviations from these assumptions? Specifically, can the authors provide more insights or alternative approaches for cases where these assumptions might not hold, such as in networks involving infinite-dimensional groups or non-compact groups?

---

> ### Author Rebuttal · Authors · 2024-08-02
>
> We appreciate your taking the time and detailed comments and suggestions.
>
> - Q1. *...How feasible is it to implement these theoretical constructs in practical neural network architectures? Have the authors considered the computational complexity and resource requirements for applying these methods to real-world datasets, and if so, what strategies or approximations can be employed to make this approach computationally efficient?*
>
> Since the ridgelet transform is given by an integral expression, it is expected that sampling from the ridgelet transform by numerical integration could replace the standard learning method by loss minimization. Actually, the idea of discretizing the integral representation has a long history and can be traced back to Barron's integral representation [B]. In theory, it is known that we can show a faster discretization rate, called the Barron rate, by theoretically conducting numerical integration with convex optimization (or equivalently, kernel quadrature or quasi-Monte Carlo methods). In practice, achieving Barron's rate is not straightforward as it reduce to another loss minimization problem. However, in recent years, Yamasaki et al. [Y] developed an exponentially-efficient quantum algorithm to sample from the ridgelet transform. So the integral representation may be advantageous when the quantum computers become practical.
>
> [B] Barron, Universal approximation bounds for superpositions of a sigmoidal function, IEEE Transactions on Information Theory, 39(3):930-945, 1993.
>
> [Y] Yamasaki et al. Quantum Ridgelet Transform: Winning Lottery Ticket of Neural Networks with Quantum Computation, ICML 2023
>
> - Q2. *The main results rely on several key assumptions, such as the local compactness of the group ( G ) and the irreducibility of the unitary representation ( \pi ). How robust are the findings to deviations from these assumptions? Specifically, can the authors provide more insights or alternative approaches for cases where these assumptions might not hold, such as in networks involving infinite-dimensional groups or non-compact groups?*
>
> First of all, locally compact group (LCG) is a sufficiently large class.
> As mentioned in ll.101-102, for example, it includes any finite group, discrete group, compact group, and finite-dimensional Lie group, while it excludes infinite-dimensional Lie groups. In particular, since finite-dim Lie groups include some non-compact groups such as $GL(n)$ and $R^n$, LCG includes those non-compact groups. Additionally, LCG may be sufficiently large to act on typical input and parameter spaces. For example, finite-dim manifolds (including finite-dim vector spaces) can be realized as homogeneous spaces of finite-dim Lie groups, so LCG may be adequate. Further, as also discussed in Limitation, whether it is really necessary to consider infinite-dimensional groups should be carefully considered.
>
> Similarly, the motivation for the unbounded case is less clear. (Just to be sure, that a linear operator is "bounded" means it is "not Lipschitz continuous", and does not mean it is "literally bounded".) Since the integration operator $DNN \circ R$ is expected to be an identity map, it usually possesses much a stronger structure than just a boundedness.
>
> However, if necessary, we will outline the basic policy of relaxing those assumptions (in the hope that motivated readers will become future collaborators). First, the assumption of LCG is required for taking an invariant measure. So, in cases of larger groups where an invariant measure cannot be taken, introducing convergence factor (an auxiliary weight function such as Gaussian) is the basic measure. Next, when the integration $DNN \circ R$ diverges, simply restricting the range of $R$ can settle the problem. This technique is adopted in Sonoda et al. [29] to show the $L^2$-boundedness.
>
> - W1. *While the paper is strong in its theoretical contributions, it lacks empirical validation through experiments or simulations...*
>
> We agree that numerical simulations are convincing. For example, we may try numerical integration of the reconstruction formula for DNNs shown in Section 4.2. However, due to the limitation of time we spent much effort on theoretical refinement. So we'd like to postpone this to important future work.
>
> - W2. *This work makes several assumptions, such as the local compactness of the group ( G ) and the boundedness of the composite operator ( \text{NN} \circ R )...*
>
> Please refer to response to Q2.
>
> - W3. *Some of the technical details, particularly those related to advanced concepts in group representation theory and the ridgelet transform, might be challenging for readers who are not experts in these areas. Providing additional intuitive explanations, diagrams, or examples to illustrate these concepts could enhance the clarity of the paper.*
>
> We appreciate your suggestions. We will add introductory explanations on group representation theory and ridgelet transform theory to the supplementary materials.

---

> ### Comment · Reviewer_bUuw · 2024-08-13
> **Re:**
>
> I thank the authors for responding to my comments. I want to keep my rating (borderline accept) since the method lacks empirical validation and is built upon various strong/weak assumptions.

---

> ### Author Response · Authors · 2024-08-14
>
> We appreciate your comment.
>
> > the method lacks empirical validation and is built upon various is built upon various strong/weak assumptions.
>
> We would like to clarify again that our assumptions are **not so strong** because
> - joint-equivariance is much a general class than equivariance,
> - locally compact group (LCG) is a sufficiently large class, and
> - the boundedness of operator $DNN \circ R$ can easily hold (note again that it is not literally bounded but just Lipschitz continuous)
>
> So, for example, our theorem covers **both shallow and deep fully-connected network** (which is **not group equivariant** but **joint-group-equivariant**) as presented in the example section
> In addition, our result trivially covers traditional group equivariant networks such as **group-equivariant convolution networks**.
>
> So, please **let us know** which specific assumptions you think are stronger.

---

### Official Review · Reviewer_joae · 2024-07-12

**Soundness:** 3
**Presentation:** 1
**Contribution:** 2
**Rating:** 4
**Confidence:** 3

**Summary:**

This work generalizes the ridgelet transform to equivariant neural networks, providing constructive proofs of universality in the general case as integrations over parameter distributions. Although such a direction had been taken up in prior work [33], they generalize it from scalar activations to vector activations, therefore encompassing more practical equivariant networks. The authors consider the form of the ridgelet transform for deep networks, and groups including the affine group and orthogonal group.

**Strengths:**

The authors provide a constructive universal approximation result, which is in contrast to many non-constructive universality results. They strictly improve on the past work of Sonoda et al [33] by extending from scalars to vectors, which is more realistic. They consider the implications of their framework on depth separations for equivariant networks.

**Weaknesses:**

Significance/novelty: The novelty relative to Sonoda et al [33] is limited, and the significance of this work to the universality and equivariance literatures is unclear. For example, many universality results already exist in equivariance (see e.g. work by Yarotsky [3], by Dym et al [2], etc.) — it is not clear how much value this extension of the ridgelet transform adds.


Clarity: I found the writing of the paper extremely hard to follow. It did not provide sufficient background on the ridgelet transform, universality results for equivariant networks (whether constructive or non-constructive), or perhaps most importantly, motivation for why one should value constructive approximation theorems for equivariant networks. It felt that one had to have read the previous works by Sonoda et al, in order to grasp why this work was important or where its novelty was, such as how vector-valued equivariant feature maps are superior to scalar-valued feature maps, what exactly formal networks are, what the practical use or theoretical value of the ridgelet transform is, etc. The work would also benefit from an outline of the sections earlier in the paper, and a more concise and early statement of what the authors consider their main theorem/s. It was not clear what the central result about the ridgelet transform was, as the transform seemed to still involve an integral in all equivariant cases, without simplification.

As a demonstration of the power of their theoretical formulation, the authors claim to show a depth separation, in which some class of networks is exponentially wide when shallow (constant number of layers), and only linearly wide when deep (linear number of layers). However, it is not clear whether they show that any shallow network is exponentially wide when representing a given function, or just the one constructed by the ridgelet transform — is this a strict depth separation?

Mathematical rigor: Although I did not check all of the math, some glaring errors stood out to me. First, the proof of Lemma 5 begins with, “Recall that a tensor product of irreducible representations is irreducible.” This is incorrect — for example, the tensor product of the irreps of the group of 3D rotations, SO(3), are reducible, and the irreps that appear in the decomposition of their tensor products are famously given by the Clebsch-Gordan coefficients (see e.g. [1]). Moreover, in the limitations section (6.1), the authors discuss the assumption that the group is locally compact, but say that this “excludes infinite-dimensional groups”. Yet, this is also false: for example, the infinite group SO(3) is compact (and therefore locally compact). In fact, several of the authors’ examples pertain to infinite groups, such as the affine group. These errors are surprising.

Also, the mathematical techniques themselves do not appear to be novel (for instance, Schur’s lemma is quite standard, and the proofs included in the main body are rather simple — Lemmas 1 and 2 are in fact widely known), and there are no experiments or practical implications, so the merit of the paper must rest on the significance of the results themselves. Unfortunately, the the broader significance of the results are not clearly demonstrated. The authors claim to reveal “the close relationship between machine learning theory and modern algebra,” but the mathematical tools they use seem like the standard ones used already throughout the equivariance literature. I am not sure what the “major upgrade to machine learning theory from the perspective of modern algebra” will therefore be.

[1] Clebsch-Gordan Nets: a Fully Fourier Space Spherical Convolutional Neural Network by Kondor, Lin, and Trivedi 2018

[2] On the Universality of Rotation Equivariant Point Cloud Networks by Nadav Dam and Haggai Maron 2020

[3] Universal approximations of invariant maps by neural networks by Dmitry Yarotsky 2018

**Questions:**

1. Do the techniques in this paper enabling proving universality of any networks for which universality (even non-constructive) was not already known? If so, it would be great to highlight these cases.

2. Can the authors clarify the two mathematical errors pointed out in the previous section (on the tensor product of irreps, and on local compactness)?

3. Is the depth separation a strict depth separation, I.e. is it the case that every shallow network representing the hypothetical function has to be exponentially wide?

4. Is there intuition for why semi-direct product arises in formal deep networks? And intuition for how formal deep networks differ from standard neural networks?

**Limitations:**

Yes, the authors discussed limitations of their work.

---

> ### Author Rebuttal · Authors · 2024-08-02
>
> We appreciate your taking the time and detailed comments and suggestions.
> We are grateful for crediting the **strict improvement from the previous study**.
>
> Let us point out and correct some major misunderstandings.
>
> - *Summary: This work generalizes the ridgelet transform to equivariant neural networks*
>
> This may cause the reviewer misevaluation. Our objective is **not "equivariant" networks but "joint-equivariant" networks**, which encompasses **both equivariant and non-equivariant maps**. For example, we deal with fully-connected networks, which is **not equivariant** in the ordinary sense. We present a unified criteria to verify the universality of deep/shallow equivariant/non-equivariant networks, which is the irreducibility of induced representation $\pi$.
>
> - *Significance/novelty: ... For example, many universality results already exist in equivariance (see e.g. work by Yarotsky [3], by Dym et al [2], etc.)*
>
> Our main result is much more general than previous studies in the universality of NNs. Typical previous studies such as Yarotsky [3] and Dym [2] are limited to specific groups $G$ (eg. compact gp, roto-translation gp $SE(n)$, symmetric gp $\mathfrak{S}_n$) acting on the Euclidean space $X=\mathbb{R}^d$, and carefully hand-crafted network architectures, while our results cover **any** locally compact group $G$ acting on **any** data domain $X$ (eg. function space, discrete sets), and **any** joint-equivariant feature map.
>
> - *Mathematical rigor: ... “Recall that a tensor product of irreducible representations is irreducible.” This is incorrect*
>
> This is incorrect. First, (a) an **external tensor product representation of irreducible representations is irreducible** (see e.g., *Folland [10, Theorem 7.12]*). On the other hand, (b) an **internal** tensor product of irreducible representations is not necessarily irreducible. It seems that the reviewer is confusing (a) with (b). In other words, the tensor product $\pi_1 \otimes \pi_2$ of the irreducible representations $\pi_1, \pi_2$ of groups $G_1$ and $G_2$, respectively, is an irreducible representation of the product group $G_1 \times G_2$ by (a), whereas it is not necessarily an irreducible representation of the component groups $G_1$ or $G_2$ by (b). In the case of Lemma 5, $\pi_1$ is irrep of $G_1 = O(m)$ on $R^m$, $\pi_2$ is irrep of $G_2 = Aff(m)$ on $L^2(R^m)$, and the representation $\pi$ in question is the tensor product $\pi_1 \otimes \pi_2$ of $G_1 \times G_2$. So, it is irreducible by (a).
>
> - *(contd) ... the authors discuss the assumption that the group is locally compact, but say that this “excludes infinite-dimensional groups”. Yet, this is also false...These errors are surprising.*
>
> This is incorrect. SO(3) is an infinite group (i.e., the cardinality of SO(3) as a set is infinite), but it is **not an infinite-dimensional group but a finite-dimensional Lie group**, and thus, it naturally falls under the category of locally compact groups.
>
> - *Also, the mathematical techniques themselves do not appear to be novel (for instance, Schur’s lemma is quite standard, and the proofs included in the main body are rather simple — Lemmas 1 and 2 are in fact widely known)...*
>
> We do not agree this. Lemmas 1 and 2  are key properties of joint-equivariant maps, which cannot be "well-known". We skimmed 50+ universality papers, but **no paper used Schur's lemma** to show the universality. **Let us know** if such a paper exist.
>
> ---
> Q1. Yes. Please refer to A.1 in the supplementary pdf. We present a new network for which the universality was not known.
>
> Q2. Neither of mathematical concerns raised are incorrect.
>
> Q3. Please refer to A.2 in the supplementary pdf. We present a clarification of depth-separation with a cyclic group.
>
> Q4. Because the semi-direct product is a sufficient condition for the equality $|G \ltimes H| = |G||H|$ holds, which maximize the effect of depth-separation.
>
> ---
> Misc.
>
> - *Clarity:...*
>
> We appreciate productive feedbacks. We add backgrounds on the ridgelet transform and literature overview on the universality NNs.
>
> - (contd) *motivation for why one should value constructive approximation theorems for equivariant networks...*
>
> In the introduction, the motivation for focusing on constructive approximation via ridgelet transform is because it can explain how the parameters of the neural network are distributed. The reason why the scalar value is insufficient is explained in the paragraph on line 43. As for the FDN, it is explained in Sec. 5, so there is no need to refer to the previous study.
>
> - (contd) *what the practical use or theoretical value of the ridgelet transform is*
>
> Since the ridgelet transform is given in closed form, finite neural networks can be obtained by discretizing it. This is not possible with non-constructive proofs. Even with constructive proofs, a carefully handcrafted network is common, which is only a particular solution to the equation $DNN[\gamma] = f$. Although not demonstrated in this study, classical ridgelet transforms can describe the general solution [29]. Therefore, any solution obtained through deep learning can be described by the ridgelet transform.

---

> > ### Comment · Reviewer_joae · 2024-08-13
> > **Thanks for the response**
> >
> > Thanks to the authors for their response. A few responses to their comments:
> >
> > 1.  Indeed, I misunderstood that this paper's results encompass not just equivariant networks, but "joint-equivariant" networks. Thank you for clarifying this. In a future draft, it would be helpful to make this distinction more clear with concrete examples of architectures from the beginning. With that said, there are even more universality results for non-equivariant networks, such as fully-connected networks. A.1 in the new PDF is helpful, as an example for which universality was apparently not known before (note that I have not verified this). I appreciate that a unified method is being used to prove universality for many types of networks at once, although I think this misunderstanding also speaks to a general lack of concrete, grounded examples and practically-motivated interpretations in the paper.
> >
> > 2. Regarding the apparent mathematical errors I mentioned, I now understand the authors' intent, but the phrasing of both statements should be updated so that they are correct in isolation. E.g., I would recommend "Recall that a tensor product of irreducible representations is irreducible." be changed to "Recall that a tensor product of irreducible representations is irreducible *in the product group*", and that they explicitly write "infinite-dimensional *Lie* group".
> >
> > 3. Lemma 1 is standard in the canonicalization literature, and simply says that a canonicalized function ($\phi(x,g)$) is equivariant. Lemma 2 is only a very minor modification of the well-known result that a composition of equivariant layers remains equivariant. Schur's Lemma is used often in equivariant universality papers (e.g. "On the Generalization of Equivariance and Convolution in Neural Networks to the Action of Compact Groups" by Kondor and Trivedi 2018), although I acknowledge not in this precise way (and not for non-equivariant networks, as far as I know).
> >
> > Overall, I still have major concerns regarding both the significance of the paper's core theoretical contribution and the paper's clarity. However, I have adjusted my score upward to a "borderline reject" in light of the authors' response to some of my criticisms.

---

> > > ### Author Response · Authors · 2024-08-14
> > >
> > > Thank you for your detailed feedback. We will update the draft according to your suggestions.
> > >
> > > Let us repeat the significance, that is, the uniformity and comprehensiveness of the main theorem. By using our main theorem, we can show the reconstruction formula (which is much stronger than just a universality) of a variety of both deep and shallow networks in a unified, constructive, and systematic manner. As we have repeatedly emphasized, the coverage is much larger than previous studies. Besides, the proof is simple by using the Schur's lemma.

---

### Official Review · Reviewer_9wR6 · 2024-07-31

**Soundness:** 4
**Presentation:** 3
**Contribution:** 2
**Rating:** 6
**Confidence:** 3

**Summary:**

The authors present a generalization of the work by Sonoda et al. by extending their formulation of universal approximation theorems applicable to a specific class of neural networks namely scalar-valued joint-group-invariant feature maps for "formal deep network" to a much larger class of learning machines. Their theory using tools from group representation theory allows them to uniformly treat both shallow and deep neural networks with a larger class of activation functions. They provide an explicit construction for parameter assignment (aka Ridgelet Transform) and apply it to vector valued joint group-equivariant feature maps.

**Strengths:**

- The topic is well motivated and the writing is clear and understandable. The interspersed explanations in plain english are quite helpful in understanding a paper that leans quite heavily on sophisticated mathematical formalisms. (eg line 93-94).
- The proofs and the notation are clear and succinct.
- The authors extend an earlier work to a much more practical and real world class of NNs by introducing *vector-valued joint group-equivariant* feature maps, which yields universal approximation theorems as corollaries. They also unify the treatment of both shallow and deep networks by leveraging Schur's Lemma.
- They provide explicit examples for depth 2 and depth $n$ fully connected network with an arbitrary activation in Section 4.2 which helps ground their method and significantly helps the reader understand how to leverage the tooling introduced by the authors.
- The paper provides formal support for the popular interpretation for the efficacy of DNNs compared to shallow networks, namely that they construct hierarchical representations which would take an exponential number of neurons to represent using a single layer.
- The limitations section is well written and is explicit about the assumptions made so that the reader is aware of the regime in which the proofs are applicable.

**Weaknesses:**

**Major**
- The biggest weakness of the work seems to be that it shares a vast amount of technical analysis, machinery and the fundamental proofs are shared with the earlier work by Sonoda et al. While the extension to a larger class of networks and the introduced vector values feature maps is certainly valuable, I am not fully convinced of the differential novelty of the work. Most of the (valuable) effort has been spent in a mostly natural extension of the previous work on the topic.


**Minor**
-  The authors mention that assumption (5) (that the network is given by the integral representation) in limitations is potentially an "advantage". If that is so, a discretized version would be the preferred model since it is also closer to real world NNs
- Typo on line 77  - mathmatical -> mathematical
- Typo on line 310 - cc-universaity -> cc-universality
- lines 135 - 137 would be significantly easier to read when broken into multiple lines

**Questions:**

- The authors allude to cc-universality in the Limitations section, can you briefly explain the term and is it the same as defined in Micchelli et al., 2006 etc
- See major weakness

**Limitations:**

No limitations.

---

> ### Author Rebuttal · Authors · 2024-08-02
>
> We appreciate your taking the time and detailed comments and suggestions.
>
> - Q1.  *The authors allude to cc-universality in the Limitations section, can you briefly explain the term and is it the same as defined in Micchelli et al., 2006 etc*
>
> Yes, it is the same. We supplement with a brief explanation in the revised version. Below is a vector-valued version.
>
> **Definition.** Let $X$ be a set, $Y$ be a Banach space, and let $F$ denote a collection of $Y$-valued functions on $X$. We say $F$ is *cc-universal* when the following condition holds: For any compact subset $K$ of entire domain $X$, any function $f$ in $F$, and any positive number $\epsilon$, there exists a $Y$-valued continuous function $g$ on $K$ such that $\sup_{x \in K} \\| f(x) - g(x) \\|_Y \le \epsilon$.
>
> It seems that the term cc-universal was introduced in the context of kernel methods in the 2010s to distinguish it from other topologies such as $c_0$-universal and $L^p$-universal. See, eg.,
>   - Sriperumbudur et al. On the relation between universality, characteristic kernels and RKHS embedding of measures, AISTATS2010.
>
> In the standard mathematical terminology, it is called *density in compact-open topology* (the topology associated with *compact convergence*).
> In the 1980s, NN researchers (such as Cybenko and Hornik et al.) called it the universal approximation property, but this terminology is ambiguous in the choice of topology, so we call it cc-universal.
>
> - Q2/W1. *The biggest weakness of the work seems to be that it shares a vast amount of technical analysis, machinery and the fundamental proofs are shared with the earlier work by Sonoda et al. While the extension to a larger class of networks and the introduced vector values feature maps is certainly valuable, I am not fully convinced of the differential novelty of the work. Most of the (valuable) effort has been spent in a mostly natural extension of the previous work on the topic.*
>
> At the state of the previous study, **it was not possible** to include **deep fully-connected** networks, and this study **resolved this issue**. It is mainly because the previous study assumes **scalar-valued maps**. To deal with deep networks, scalar-valued is insufficient not only because typical hidden layer maps are vector-valued, but also because just taking a tensor-product of joint-invariant scalar-valued maps (which is vector-valued but) in general cannot represent a joint-equivariant vector-valued map. Without joint-equivariance, we cannot fully investigate the effect of group action on hidden layers (such as $NN_2$ in $NN_3 \circ NN_2 \circ NN_1$). So, we need to rebuild the framework from scratch.
>
> Technically, we replaced the scalar-valued joint-invariant maps with *joint-equivariant maps between any $G$-sets* (Definition 3 is much more general than vector-valued maps). As described after Definition 3, this replacement resolves several technical issues. In particular, it could naturally deal with function composition (Lemma 2). As a result, we succeeded to deal with deep networks.
>
> Additionally, another technical difficulty occurs in applying the main theorem, that is, to **find an irreducible representation**. In the end, of course, we have discovered the one (**Lemma 5**).
>
> Note also there are many formulations we tried but did not work out before we arrived at this proof. Therefore, please be aware that what seems to be an obvious generalization is a kind of so-called *Columbus' egg* illusion.
>
> - W2. *The authors mention that assumption (5) (that the network is given by the integral representation) in limitations is potentially an "advantage". If that is so, a discretized version would be the preferred model since it is also closer to real world NNs*
>
> We agree your suggestion, and tried to write down the $cc$-universality of finite DNNs during the rebuttal period. However, the proof gets more than 3 pages. So we'd like to postpone to our important future work. The proof essentially repeats a parallel argument with Appendix A.2 in [30] twice: One is for discretizing single hidden layer (eq.24), the other is for discretizing entire network (eq.25). Both convergence are justified by the dominated convergence theorem for the Bochner integral.

---

> > ### Comment · Reviewer_9wR6 · 2024-08-11
> >
> > Thank you for your detailed response. I would like to increase my score based on your responses to my questions (and reviewer joae). While numerical simulations would certainly be helpful to ground the method (as mentioned by Reviewer bUuw), I am convinced that the theoretical contributions are worthy enough to be published now.
> >
> > It would be very helpful for readers to *add example A.2 from the global response to the paper/appendix*.

---

> > > ### Author Response · Authors · 2024-08-14
> > >
> > > Thank you for your response. We'll update our draft according to your suggestions.
> > >
> > > ---
> > > Related to A.2, we'd like to further point out that the depth-separation has an effect on the generalization error bound.
> > > For example, in a simple case when the hypothesis class $\mathcal{F}$ is parameterized by a finite set, say $\Theta$, then the generalization error is upper bounded by the cardinality $|\Xi|$ of the parameter set in the form:
> > >
> > > (expected risk) $\le$ (empirical risk) + $c \sqrt{\log |\Theta|/n}$
> > >
> > > with some constant $c$. The proof is a consequence of the so-called Massart's Lemma.
> > >
> > > Since the expressive power of NN1 and NN2 are the same, both network can achieve the same empirical risk.
> > > Nonetheless, the variance term $\sqrt{|\Theta|/n}$ for depth-2 network NN2 is **exponentially smaller** than the one for depth-1 network NN1 because $|\Theta|$ is given by a *sum* $|C_2| + |C_3^2| = 5$ for NN2 while it is given by a *product* $|C_2||C_3^2|=6$ for NN1.
> > >
> > > When the parameter space $\Theta$ is not a finite set but a finite-dimensional bounded domain,
> > > the variance term is given by its dimension: $\sqrt{\dim \Theta/n}$ (the proof is given by the metric entropy arguments),
> > > and the similar arguments hold because the dimensions of parameter space $\Theta$ are given by a *sum* $|C_2| + |C_3^2| = 5$ for NN2 while it is given by a *product* $|C_2||C_3^2|=6$ for NN1.

---

### Author Rebuttal · Authors · 2024-08-07

We thank the reviewers for their valuable comments and detailed questions. In response to several questions, we have supplemented two additional examples.

- In A.1, we present a new network for which the universality was not known.
- In A.2, we present a clarification of depth-separation.

---

### Decision · Program_Chairs · 2024-09-25

**Decision:**

Reject

**Comment:**

Thank you for your valuable contribution to Neurips and the ML community. Your submitted paper has undergone a rigorous review process, and I have carefully read and considered the feedback provided by the reviewers as well as the private messages to AC.

The work introduces a unified framework for universal approximation theorems for neural networks, utilizing group representation theory. It broadens the scope to include vector-valued joint-group-equivariant feature maps, offering a systematic approach for both shallow and deep neural networks with nonlinear activation functions. Using Schur's lemma, the paper demonstrates that these networks can universally approximate any function within a specific class.

The paper received borderline final review scores (4,5,6). Certain critical issues were raised including  (i) significance of the core contribution with respect to prior work (ii) limited novelty in the proof technique (iii) lack of clarity, (iv) lack of empirical results. While the lack of experimental validation is understandable for theory papers, the reviewers agree that the novelty in the proof technique is somewhat limited and the clarity needs to be improved. In the rebuttal, the authors addressed some of these concerns; however, their rebuttal was not sufficiently compelling to the reviewers. The discussion with Reviewer joae clarified some issues but their final score was 4 (borderline reject).

Regarding the significance of contributions, Reviewer 9wR6 noted that much of the technical analysis, machinery, and fundamental proofs in this work are shared with earlier work by Sonoda et al., raising concerns about the novelty of the contributions. While the rebuttal successfully addressed some of the confusion, a major revision is needed to clearly highlight the novel aspects of this work.

Regarding clarity, Reviewer bUuw mentioned that advanced concepts in group representation theory and the ridgelet transform, might be challenging for readers who are not experts in these areas, and suggested that providing additional intuitive explanations, diagrams, or examples to illustrate these concepts could enhance the clarity of the paper.

Given the current form of the paper, I regret to inform you that I am unable to recommend the acceptance of the paper for publication at Neurips. I want to emphasize that this decision should not be viewed as a discouragement. In fact, the reviewers and I believe that your work has quite novel theory, valuable insights and, with further development and refinement, can make a meaningful impact on the field.

I encourage you to carefully address the feedback provided by the reviewers and consider resubmitting the paper. Please use the comments and suggestions in the reviews to improve and refine your work.
Best,
AC